# Mechanical Thrombectomy in Cerebral Venous Sinus Thrombosis: Reports of a Retrospective Single-Center Study

**DOI:** 10.3390/jcm11216381

**Published:** 2022-10-28

**Authors:** Farzaneh Jedi, Gero Dethlefs, Till-Karsten Hauser, Florian Hennersdorf, Annerose Mengel, Ulrike Ernemann, Benjamin Bender

**Affiliations:** 1Department of Diagnostic and Interventional Neuroradiology, University Hospital Tübingen, 72076 Tübingen, Germany; 2Department of Neurology and Stroke, Hertie-Institute for Clinical Brain Research, University Hospital Tübingen, 72076 Tübingen, Germany

**Keywords:** cerebral venous sinus thrombosis, mechanical thrombectomy, endovascular therapy, thrombolysis

## Abstract

Current standard care for acute cerebral venous sinus thrombosis (CVST) includes either intravenous heparin or subcutaneous low-molecular-weight heparin, but patients with refractory CVST, despite adequate anticoagulation therapy, may benefit from mechanical thrombectomy (MT). A retrospective study of patients with CVST, who underwent MT between 2011 and 2019, was performed looking at procedure success rate and clinical outcomes. Two raters evaluated the cerebral venous system of every patient before and after the intervention using the following scoring system: (0) No obvious thrombosis; (1) thrombosis without impaired blood flow; (2) thrombosis with impaired blood flow; (3) and thrombosis with complete vascular occlusion. The success of MT was measured using a score quotient (Q = A/B), dividing the sum of the patient’s scores after the intervention (A) by the sum of scores before the intervention (B). Overall, 21 MTs were performed on 20 patients with refractory or severe CVST. Clinical improvement was seen in 61.9% during hospital stay and in 80% at 6-month follow-up, with complete recovery in 70% of patients. Patients with favorable outcomes had significantly lower recanalization quotients (*p* = 0.008). Our study provides evidence supporting that MT may be a safe and effective treatment with favorable clinical outcomes for selected patients with CVST.

## 1. Introduction

Cerebral venous sinus thrombosis (CVST) is an uncommon cause of stroke (0.5–1%) [1,2]. It most commonly affects women and younger age groups. Clinical presentation varies from headache to focal neurological deficits, altered mental status, seizure and coma, which makes it challenging to diagnose and manage [3,4,5,6,7,8]. Acquired and genetic predisposing risk factors are presumed to be the causes of hypercoagulable state in CVST such as pregnancy, drugs with prothrombotic action, malignancies, dehydration, infections and systemic inflammatory diseases, trauma, thrombophilia, and other coagulopathies [5,6,7,8].

Poor neurological outcomes, in terms of death or dependency, are seen in approximately 13% even after treatment with anticoagulation [3]. Risk factors for poor prognosis include male sex, age >37 years, coma, impaired mental status, intracranial hemorrhage (ICH) on admission, thrombosis of the deep cerebral venous system, central nervous system infection, cancer and posterior fossa involvement [3,4,7,9]. The most frequent cause of death according to the International Study on Cerebral Vein and Dural Sinus Thrombosis (ISCVT) was transtentorial herniation, due to mass effect of either a focal lesion or multiple (hemorrhagic) lesions and edema [9].

Occlusion of a venous sinus leads to an increase in intravenous pressure, the arterial to venous gradient decreases, which results in decreased cerebral blood flow and brain tissue oxygenation. Cytotoxic edema followed by venous infarction, congestive hemorrhagic conversion and increased intracranial pressure account for mass effect and varying degrees of clinical presentations [5,10]. The reestablishment of venous drainage by either complete or partial recanalization of occluded vessels prevents the progress and complications and improves the outcome in comparison to patients with no recanalization [11].

The current standard of care for acute CVST even those with ICH includes either intravenous heparin or subcutaneous low-molecular-weight heparin [12,13,14]. Decompressive surgery is recommended for patients with parenchymal lesions impending herniation [12,13,14,15].

Patients with poor prognosis, those with clinical deterioration despite anticoagulation therapy and patients with contraindications for anticoagulation therapy may benefit from early endovascular interventions such as intrasinus thrombolysis (IST), or mechanical thrombectomy (MT) [3,13,14,15,16,17].

Evidence is lacking about efficacy and safety of using MT in patients suffering CVST, as most published data come from case reports or small case series since its first performance in the 1990s. Only one prospective randomized trial has been performed, that showed no difference in functional outcome [18]. This study had at least two major limitations: First considering only the rare occurrence of CVST in patients with risk factors, the study had to be stopped preliminary because of futility after inclusion of 67 patients and secondly interventions included “aggressive” techniques like using AngioJet^®^, that leads to an increase in venous pressure, or local application of alteplase or urokinase that may be associated with a poor outcome [4]. The aim of our study was to gain a better understanding of MT results on clinical improvement in patients with severe CVST.

## 2. Materials and Methods

The study was approved by the institutional ethics committee (Faculty of Medicine, University Hospital Tübingen, approval code: 573/2018BO2). Informed consent was waived in accordance with local data protection laws, due to the retrospective nature and the anonymized evaluation.

All patients with CVST that underwent interventional treatment between January 2011 and December 2019 were identified by searching our radiological information system. Available clinical and radiological data of those undergoing MT in our department were analyzed.

### 2.1. Data Retrieval

Demographic and clinical data, risk factors, details of systemic anticoagulation therapy, data on outcomes and periprocedural complications, degree of neurological deficit at follow-up, and death were collected from the electronic medical files of the patients. Presenting signs and symptoms were categorized into (1) headache, (2) focal neurological deficit, (3) altered mental status or coma, and (4) seizure. The diagnosis of CVST was made on the basis of clinical presentation, CT venography or MRI including venography and was confirmed with cerebral angiography directly before MT. Severity of clinical symptoms was evaluated by modified Rankin Scale (mRS) and severity of venous involvement by extension of thrombosis on imaging. Laboratory studies were performed to find the causes and risk factors of CVST.

Imaging data (number and extent of sinuses involved, presence and type of ICH, venous infarcts, edema and mass effect), procedural method and devices and degree of recanalization were collected from our Picture Archiving and Communication System.

For scoring recanalization rate two board certified radiologists (7 and 11 years of experience) evaluated all available imaging modalities independently. The cerebral venous system was divided into 10 anatomical regions (see Figure 1) which were evaluated individually before and after the intervention using the following scoring system:

0: No obvious thrombosis/anatomical variability such as absence or hypoplasia of a sinus;1: Thrombosis without impaired blood flow;2: Thrombosis with impaired blood flow;3: Thrombosis with complete vascular occlusion.

**Figure 1 jcm-11-06381-f001:**
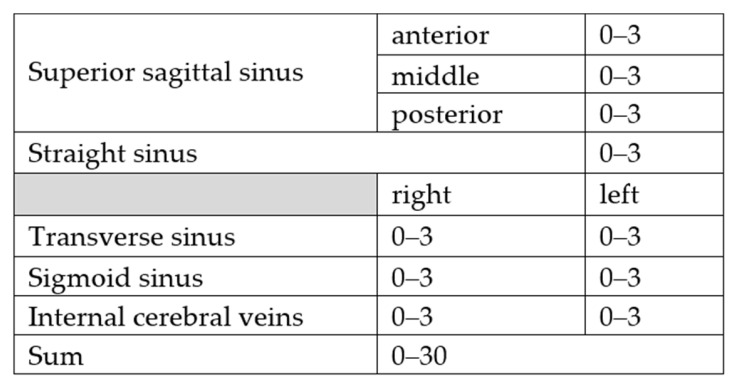
Cerebral venous system divided into 10 anatomical regions.

The recanalization success rate was calculated by a simple quotient (Q), dividing the sum of the patient’s score after the intervention (A) by the sum of score before intervention (B). A Q of 0 means a complete recanalization of all previously occluded vessels, a Q of 1 means no difference in occluded venous vessels before and after intervention and a Q > 1 reflects a worsening of sinus occlusion during intervention.

At discharge and at 6-month follow-up, mRS was used to assess clinical outcome and evaluate functional recovery. Good clinical outcome was defined as a mRS of 0–2 at 6 months follow-up. If no 6 months follow-up data were available, mRS at discharge was used alternatively.

All pseudonymised information was recorded in a spreadsheet. Statistical evaluation was performed on anonymized data after completion of data retrieval and removing all identifying information.

### 2.2. Statistical Evaluation

Statistical evaluation was performed with SPSS (Version 24, IBM, Armonk, NY, USA). Inter-rater reliability was evaluated by calculating a linear weighted kappa and the Pearson correlation coefficient between the raters. Agreement was considered as slight (kappa 0–0.21), fair (kappa 0.21–0.4), moderate (0.21–0.6), substantial (0.671–0.8) or excellent (>0.8). Mean Q scores of both raters were compared using a two-sided Student’s *t*-test, after testing for normal distribution with the Shapiro–Wilk test and variance homogeneity with Leven test. An alpha level of <0.05 was considered to be significant.

### 2.3. Venous Mechanical Thrombectomy

The decision to perform endovascular intervention was based on a case-by-case evaluation in an interdisciplinary decision-making process between neurologists and interventional neuroradiologists and was limited to cases with severe or refractory CVST. Refractory CVST was defined as deterioration of clinical status despite medical management, while severe CVST was defined as extensive involvement of cerebral venous system, raised intracranial pressure due to mass effect from a venous infarction or ICH, rapid progression of neurological deficits or mental status impairment and coma.

The approach of venous recanalization at our institution is operator dependent and has changed over time with availability of new devices, but the basic approach can be described as follows: Without stopping the intravenous heparin drip, transfemoral arterial access is established using a 4F short arterial sheath and catheterization of the internal carotid artery ipsilateral to the occluded sinus is obtained to perform diagnostic cerebral angiograms with attention to the venous phase to delineate the extent of the venous sinus involvement (see Figure 2).

Transfemoral venous access is established to perform MT under general anesthesia either with a 6 F/7F long sheath or a guiding catheter placed distally in the internal jugular vein/jugular bulb or sigmoid. A microcatheter was navigated over a microwire into the occluded venous compartment (all sinuses are accessible). The MT methods include: direct aspiration (with a VacLok syringe), stent retriever thrombectomy (retrieving the clot with a 6 × 30 mm Solitaire or Catch Maxi stent), balloon-assisted MT and venoplasty or a combination of these methods.

A control CT venography is performed after the intervention to evaluate sinus patency and follow-up complications and course of thrombosis changes.

## 3. Results

A total of 20 patients were included in the study. One patient required a second procedure within 6 days, because of poor recanalization success and further clinical deterioration after the first MT. Of the 20 patients, 12 (60%) were female; and mean age was 36 years. Table 1 outlines the age range, etiological risk factors and presenting signs and symptoms. In traumatic cases a skull fracture crossed a transverse sinus or the sigmoid sinus and caused traumatic sinus thrombosis. In all three of these traumatic cases traumatic brain injury and traumatic hematoma were also present.

Intraparenchymal hemorrhage was seen in nine patients, nine patients had venous infarction, twelve patients showed subarachnoid hemorrhage and five had associated subdural hemorrhage. An intracranial pressure (ICP) monitoring device was inserted in seven patients, and external ventricular drainage was performed in two patients with pathologically elevated ICP. Out of these patients, five required decompressive hemicraniectomy because of elevated ICP. Table 2 shows the imaging findings on admission and after MT.

All patients had been fully anticoagulated with systemic intravenous unfractionated heparin as initial treatment as soon as the diagnosis was confirmed; even in presence of ICH (target PTT ≥ 40 s or twice the normal pretreatment values). None received intravenous or local endovascular thrombolysis before MT. The mean time from symptom onset to MT was 4 days (between 0–21 days), while the mean time between admission in the hospital and MT was only 2 days (between 0–7 days). Treatment was usually continued as oral anticoagulant therapy for 12 months.

In all patients, multiple sinuses and veins were involved. Overall, 115 thrombosed segments were identified in 21 cases. Forty-two thrombosed segments were in the superior sagittal sinus (36.5%), 34 in the transverse sinus (29.6%), 27 in the sigmoid sinus (23.5%), 8 in the straight sinus (7%), and 4 in the internal cerebral veins (3.5%). Inter-rater reliability was excellent with a linear weighted kappa of 0.81 and a correlation coefficient of 0.91 for all evaluated segments before and after MT.

The most frequently utilized type of endovascular intervention was using stent retrievers in 20 cases. This was carried out as a single technique in eight cases, in combination with aspiration catheters in eight cases, and combined with Balloon venoplasty without stenting in four cases. In only one case, an aspiration catheter was used. The only procedural complication was access site bleeding in the groin in one patient. Successful partial or complete recanalization of the occluded sinus was achieved in 85.7% of cases (Q < 1), while recanalization was unsuccessful in three patients (Q = 1).

Clinical improvement was achieved in 13 of 21 cases (61.9%) during hospital stay. The clinical status of three patients remained unchanged according to mRS yet they too showed a regression of symptoms on follow-up. All patients presented with favorable neurologic outcomes (mRS 0–2) at follow-up (≥6 months). At 6 months follow-up, recovery was complete (mRS 0) in nine (45%) patients, two (10%) patients had mild residual symptoms (mRS 1), and two (10%) showed moderate residual symptoms (mRS 2). Three patients were lost during follow up, of whom two had mild residual symptoms (mRS 1) and one showed moderate residual symptoms (mRS 2) at discharge. 

Five patients (23.8%) experienced clinical deterioration during hospital stay. Four died within 10 days after MT and one required a second MT with favorable neurologic outcome. The cause of death was severe traumatic brain injury in two patients, malignant media infarction (unrelated to the MT) in one patient, and severe bleeding and edema in one patient suffering from rhabdomyosarcoma.

The mean Q value was 0.7 and did not differ significantly between both raters (Rater 1 0.71 ± 0.17, Rater 2 0.68 ± 0.19, paired *t*-test, t (20) = 1.448, *p* = 0.163). Patients with a good clinical outcome had significantly better Q (0.64 ± 0.16) than patients with a poor outcome (0.88 ± 0.10) (t (19) = −2.95, *p* = 0.008) (see Figure 3).

Table 3 summarizes intervals from symptom onset to MT, MT details, outcomes and complications.

## 4. Discussion

Although anticoagulation remains the basic treatment for patients with CVST, it may not be the only choice for refractory and complicated cases with severe neurological deficits, coma or with major contraindications to anticoagulation. Refractory CVST can rapidly progress to cause ischemic and hemorrhagic strokes, cerebral edema, mass effect, and eventually death. These patients may benefit from early endovascular interventions such as IST, MT, or a combination of both [3,13,14,15,16,17]. Approximately two-thirds of patients treated with anticoagulation show recanalization, which generally takes weeks or months. However, 95% of patients undergoing MT have shown a rapid complete or partial recanalization in different case series and prospective cohorts [4].

In our study, 61.9% showed an improvement of mRS during hospital stay and 80% at 6 month follow-ups. Complete recovery rate was 70% and none of our surviving patients ended up with severe disability. Our study’s partial or complete recanalization rate was 85.7%. Thrombectomy success rate measured with the recanalization quotient Q differed significantly between patients with clinical improvement and patients with poor outcome. In other words, patients with a higher recanalization rate fully recovered more often and had less residual symptom and disabilities, which could imply that rate of recanalization could be a predictor for clinical outcome.

In ISCVT, the largest international study on CVST to date, out of 624 patients who had undergone systemic or local pharmacological treatment, 57% became symptom-free (mRS 0), 22% had minor residual symptoms (mRS 1), 8% maintained mild impairments (mRS 2), 5% remained moderately or severely impaired (mRS 3 to 5), and 8% did not survive the median follow-up of 16 months [3]. This implies that we had a comparable rate of good clinical outcome but a higher mortality rate. This may be due to the fact that in our cohort all patients had very severe symptoms or clinical worsening under standard therapy and therefore present a selected subsample of severe and refractory CVST. Furthermore, our mortality rate of 20% was mainly due to comorbidities, such as severe traumatic head injury in two patients, malignant media infarction in one, and poor health condition due to coexisting malignancy in one patient. It should be considered that these four patients were already in a critical condition with an mRS of 5 at admission. This agrees with previous studies mentioning ICH on admission and cancer as risk factors for poor prognosis [3,4,7,9].

The most important complication of MT is new ICH, reported in 10% of patients. Other rare complications may be access site hematoma, peripheral nerve injuries, soft tissue infections and iatrogenic arteriovenous fistula formation [4]. Our patients showed no ICH as a result of MT; and procedural complication occurred in only one patient in form of access site hematoma in the groin.

Most of the existing studies and reviews performed over the last 3 decades have discussed the combination of IST and MT in patients with neurological deterioration refractory to anticoagulation therapy with heterogeneous results [4,5,6,16,19,20,21,22,23,24]. A systematic review of 42 studies and 185 patients with severe CVST who were treated with MT ± IST revealed poor outcome or death (12%) despite good recanalization rates in 16% of cases [4]. Therefore, since administration of thrombolysis within the venous system may be associated with a worse outcome, none of our cases received intravenous thrombolysis. 

Studies focusing only on MT results in treatment of patients with CVST are lacking. Table 4 summarizes the available data of studies with a sample size ≥ 5 in comparison to our results [25,26,27,28,29]. Soleau et al. compared the therapeutic results on 31 cases that underwent 4 different treatment strategies including medical observation only (*n* = 5), systemic anticoagulation therapy with heparin (*n* = 8), endovascular thrombolysis with urokinase or tissue plasminogen activator (*n* = 10) and MT (*n* = 8). The patients treated with MT had the best clinical outcome with 88% clinical improvement at discharge [25]. In their study Shui et al. included patients with mild symptoms such as isolated headache or seizure (54%), which are usually not treated with MT. Additionally, the authors did not clearly define good neurological outcome; therefore, the results are hard to interpret [27]. To the best of our knowledge none of the other studies with a sample size ≥ 5, included traumatic sinus thrombosis, which might be the major reason for the smaller percentage of at least partial recanalization and the higher mortality in our cohort. Only one patient with traumatic sinus thrombosis in our cohort had a favorable outcome and a good recanalization quotient, while the other two did not survive with a Q of 0.8 and 1.

Degree of sinus occlusion has been presented using general terms such as partial or complete thrombosis in previous studies and a precise definition has not been provided. Especially the term “complete recanalization” could be misleading. While we saw a complete flow restoration in some of our patients, a complete recanalization without any signs of residual thrombus was not achieved. It remains unclear how other studies defined complete recanalization. The strength of our study was that we developed and used a scoring system to standardize and define degree of sinus occlusion and thrombectomy success rate, hereby reducing the risk of individual interpretation. This scoring system showed an excellent inter-rater reliability. In addition, we used a standard protocol for diagnosis, treatment and follow-up of patients. 

Potential weaknesses of our study include the retrospective nature and relatively small sample size, which affects the result of multivariable analyses and should be taken into account particularly when interpreting the results. A selection bias could be ruled out because all the patients who underwent MT were included in our cohort. While our data suggest that a higher recanalization rate (smaller Q) implies a better clinical outcome, traumatic sinus thrombosis was a major confounder of poor outcome. The correlation between traumatic sinus thrombosis, recanalization rate and clinical outcome could not be further evaluated due to the small sample size. Another potential limitation of our study could be the fact that we did not consider bridging and cortical veins in our evaluations; however, major symptoms of sinus vein thrombosis occur if the thrombus extends from the sinus into these small veins which MT is unable to recanalize because of the risk of perforation [30]. In addition, limitations based on a heterogeneous study cohort using different thrombectomy methods and lack of a control group may have affected our results. Especially without a control group, it remains unclear if clot stability affects both mechanical recanalization rate and response to heparin therapy. Lastly, biological results of heparin therapy (like anti-Xa activity or APTT ratio) were not available for all patients and therefore not evaluated, which could be a potential co-factor for the outcome.

## 5. Conclusions

The aim of therapy in CVST is earlier recanalization of the sinuses and thereby preventing complications. Although anticoagulation is the current standard of therapy, considering the low complication rate and the clinical and radiological success, our study provides evidence supporting the efficacy and safety of MT in refractory cases in which systemic anticoagulation fails. However, evidence is lacking about the proper indication, timing, approach and devices used in these procedures. Invention of newer flexible devices might reduce catheter-related limitations and complications such as the risk of venous sinus rupture. More multicenter academic trials with a large sample size are needed to perform multivariable regression analyses on this low incidence disease with large pathophysiological heterogeneity, in order to provide reliable data on efficacy and safety of MT to guide in clinical decision making.

## Figures and Tables

**Figure 2 jcm-11-06381-f002:**
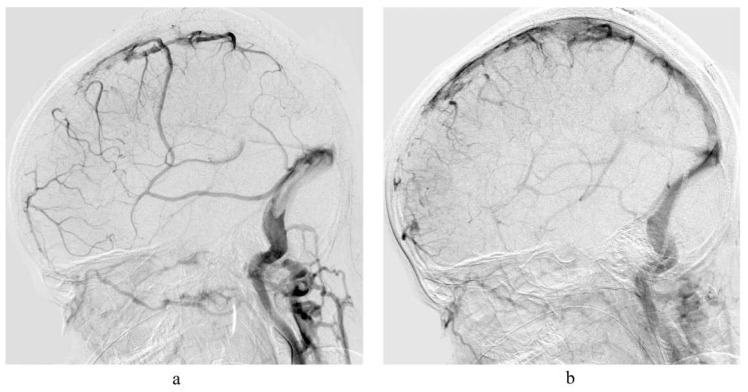
A child with a history of acute lymphocytic leukemia (ALL) presented with altered mental status. (**a**) Preprocedure lateral left internal carotid artery angiogram demonstrates thrombosis of superior sagittal sinus and left transvers sinus; (**b**) Postprocedure lateral left internal carotid artery angiogram after treatment with Solitaire Revascularization Device demonstrate patency of the superior sagittal sinus and left transverse sinus with satisfactory flow (Q = 0.57).

**Figure 3 jcm-11-06381-f003:**
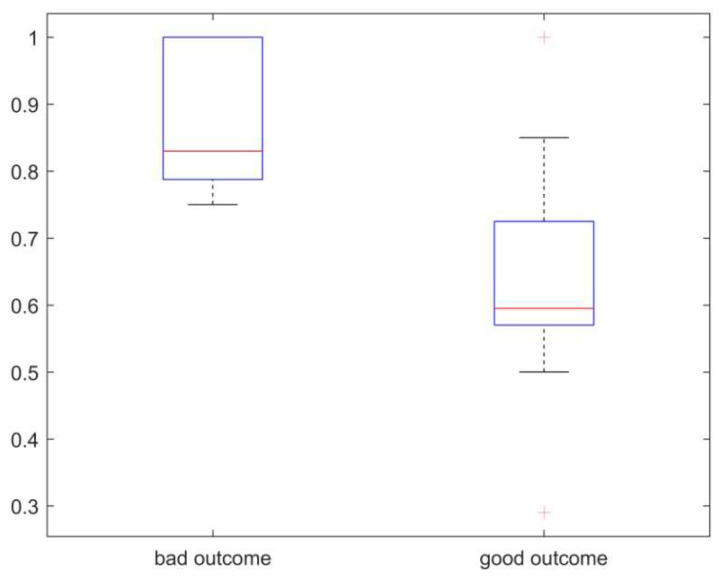
Boxplot of recanalization quotient (0 = complete recanalization, 1 = no recanalization) in procedures with bad outcome (worsening of mRS) and good outcome (improvement of mRS).

**Table 1 jcm-11-06381-t001:** Patient demographics.

Patient	Age	Sex	Presenting Symptoms	Risk Factors
1	38	F	Altered mental status	Unknown
2	22	F	Coma	Trauma
3	79	M	Coma	Trauma
4	20	F	Headache, focal neurological deficits, Seizure	Oral contraceptives
5	29	F	Headache, focal neurological deficits	Factor V Leiden
6	21	M	Coma	Trauma
7	9	M	altered mental status	Urinary tract infection
8	49	M	Headache, focal neurological deficits	Multiple sclerosis
9	58	F	Altered mental status, focal neurological deficits, Seizure	Thrombophilia of unclear cause
10	41	F	Headache, altered mental status, focal neurological deficits	Thrombophilia of unclear cause
11	52	M	Headache, focal neurological deficits	Malignancy
12	21	M	Headache, altered mental status, focal neurological deficits	Malignancy
13	50	M	Headache, focal neurological deficits	Unknown
14	14	M	Headache, altered mental status, focal neurological deficits	Unknown
15	22	F	Headache, altered mental status	Pregnancy
16	57	F	Headache, focal neurological deficits	Factor V Leiden
17	55	F	Headache, altered mental status, focal neurological deficits	Unknown
18	33	F	Headache, focal neurological deficits	Ulcerative colitis, 3 weeks postpartum
19	22	F	Headache *	Oral contraceptives, Prothrombin G20210A
20	34	F	Headache, altered mental status, Seizure	Pregnancy, mutation of Methylene
				Tetrahydrofolate Reductase

Female (F), male (M), * The decision to perform thrombectomy in this young patient was based on extended thrombosis.

**Table 2 jcm-11-06381-t002:** Imaging findings.

Patient	IPH	SDH	SAH	VI	EVD	ICPMD	DHC	Location of CVST	Findings in Follow-Up Imaging after MT till Discharge
1	+	+	+	+	-	+	before MT	SigS, SS, TS	no rebleeding, infarction constant
2	+	+	+	-	-	+	before MT	SigS, TS	increasing epidural bleeding after Craniotomy
3	+	+	+	-	+	+	-	SigS, TS	increasing IPH and edema
4	-	-	-	+	-	-	-	SSS, SigS, TS	decreasing edema, no bleeding
5	-	-	-	-	-	-	-	SSS, SigS, SS, TS	no bleeding, no infarction
6	+	+	+	-	-	+	before MT	SigS, TS	no rebleeding, no infarction
7	-	-	-	-	-	-	-	SSS, TS	no bleeding, no infarction
8	-	-	+	-	-	-	-	SSS, SigS, TS	no bleeding, no infarction
9	+	-	+	+	-	-	-	SSS, SigS, SS, TS	no rebleeding, infarction constant
10	-	-	-	*	+	+	after MT	SSS, SigS, TS	malignant media infarct with bleeding
11	+	+	+	+	-	-	-	SSS, SigS, TS	no rebleeding, infarct constant
12	+	-	-	+	-	+	after MT	SSS, SigS, TS	severe rebleeding, increasing edema
13	-	-	-	-	-	-	-	SSS, SigS, SS, TS	no bleeding, no infarction
14	-	-	-	-	-	-	-	SigS, TS	no bleeding, no infarction
15	-	-	+	+	-	-	-	SSS, SigS, TS	no rebleeding, infarction constant
16	-	-	+	-	-	-	-	SSS, SigS, TS	no rebleeding, no infarction
17	+	-	+	+	-	-	-	SigS, TS	no rebleeding, infarction constant
18	+	-	+	+	-	+	-	SSS, SigS, SS, TS, ICV	no rebleeding, infarction constant
19	-	-	-	-	-	-	-	SSS, SigS, SS, TS, ICV	no bleeding, no infarction,
20	-	-	+	+	-	-	-	SSS, SigS, SS, TS, ICV	no rebleeding, infarction constant

Intraparenchymal hemorrhage (IPH), Subdural hemorrhage (SDH), Subarachnoid hemorrhage (SAH), Venous infarctions (VI), External ventricular drainage (EVD), Intracranial pressure monitoring device (ICPMD), Decompressive hemicraniectomy (DHC), Cerebral venous sinus thrombosis (CVST), Superior Sagittal Sinus (SSS), Straight Sinus (SS), Transverse Sinus (TS), Sigmoid Sinus (SigS), Internal Cerebral Veins (ICV), + present, - not present, * Malignant middle cerebral artery infarction.

**Table 3 jcm-11-06381-t003:** Procedural information and outcomes.

Patient	Interval from Clinical Onset	Admission to MT	MT Device	Complication	B *-**	A *-**	Q	mRS at Admission	mRS at Discharge	mRS at 6m
1	4d	4d	Penumbra, Solitaire	None	7-9	5-5	0.63	5	4	1
2	4d	4d	Solitaire, Balloon-PTA	increasing epidural bleeding after Craniotomy	4-4	4-4	1	5	6	died
3	1d	1d	Solitaire	increasing IPH	5-5	4-4	0.80	5	6	died
4	3d	3d	Solitaire, Balloon-PTA	None	12-13	7-7	0.56	4	2	0
5	10d	1d	Solitaire, Balloon-PTA	None	16-17	10-10	0.61	1	1	1
6	1d	1d	Solitaire, Balloon-PTA	None	6-4	2-1	0.29	5	1	2
7	2d	2d	Solitaire	access site bleeding in groin area	13-13	7-8	0.58	3	2	NA
8	21d	7d	Solitaire	None	16-15		0.84	1	1	0
9	0d	0d	Solitaire	None	13-11		0.58	5	1	0
10	2d	2d	Solitaire	malignant media infarction with bleeding	9-11	7-8	0.75	5	6	died
11	1d	1d	Catch Maxi	None	10-9	6-5	0.58	4	4	0
12	11d	1d	Catch Maxi	None	12-11	12-11	1	5	6	died
13	0d	0d	Fargo + Catch Maxi	None	20-15	11-11	0.64	4	2	0
14	0d	0d	Solitaire	None	6-6	6-6	1	5	0	0
15	0d	0d	Catch Maxi, Sofia Plus	None	18-18	9-9	0.50	4	1	NA
16	6d	1d	Catch Maxi, Sofia Plus	None	15-16	9-8	0.55	3	1	0
17	2d	1d	Solitaire, Sofia Plus	None	6-6	4-3	0.58	3	1	0
18a	1d	0d	Catch Maxi, Sofia Plus	None	24-23	12-18	0.83	3	5	see 18b
18b	7d	6d	Catch Maxi, Sofia Plus	None	17-16	14-12	0.79	5	4	2
19	2d	0d	Sofia Plus	None	16-16	10-11	0.66	1	0	0
20	6d	0d	Catch Maxi, Sofia Plus	None	26-26	22-22	0.85	5	1	NA

Mechanical thrombectomy (MT), Sum of the patient’s score before intervention (B), Sum of the patient’s score after the intervention (A), Q = A/B, Modified Rankin Scale (mRS), * rater 1, ** rater 2, Intraparenchymal hemorrhage (IPH), Not Available (NA).

**Table 4 jcm-11-06381-t004:** Previous studies.

Study	N (Follow-Up)/Interventions	MT Method	At Least Partial Recanalization	Good Outcome (mRS 0–2)	Complications	Mortality
Soleau et al. [25]	8/8	B	6 (75%)	4 (50%)	2	1
Dashti et al. [26]	13 (9)/13	AngioJet	13 (100%)	7 (53.8%)	NA	2
Shui et al. [27]	26/26	B	26 (100%)	26 * (100%)	0	0
Ma et al. [28]	23/23	S	23 (100%)	22 (95.6%)	0	0
Styczen et al. [29]	13/14	A/A+S	13 (92%)	12 (85.7%)	3	1
this study	20/21	BP+S/A/S/A+S	18 (85.7%)	17 (80.9%)	1	4

* Clinical outcome only described as favorable, modified Rankin Scale (mRS), Balloon MT (B), Stent-retriever (S), Aspiration catheter (A), Balloon venoplasty (BP), Not available (NA).

## Data Availability

The anonymized raw data presented in this study are available upon reasonable request from the corresponding author.

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
