# Peer review of "Mechanical Thrombectomy in Cerebral Venous Sinus Thrombosis: Reports of a Retrospective Single-Center Study"

_jcm, 2022, doi:10.3390/jcm11216381_

Round 1

Reviewer 1 Report

I read with interest the paper entitled: "mechanical thrombectomy in cerebral venous sinus thrombosis: reports of a retrospective single-center study".
It is a retrospective, monocentric study with a small number of patients, but with a selection of patients by the severity of the clinical picture.

The clinical and radiological data are precise, but I would like to know more about the reason for the decision to perform a thrombectomy.

- the 20 patients all have a severe CVST or refractory to treatment but a few patients had early thrombectomy (average delay: 4 days ; 0 to 21 days). What event decided the procedure in these patients with a severe initial picture ?                                                                              Thrombectomy in stroke should be performed within the first few hours; why should this instruction not be valid for CVST ?                  Isn't there a risk of reduced efficacy if the procedure is delayed ?      How many patients had the thrombectomy on the day of diagnosis (D0), what was the outcome for these patients ?

- regarding the anticoagulant treatment, the data are insufficient: type of heparin (UFH and/or LMWH), dosage, biological results (APTT ratio, anti-Xa activity)? definition of resistance to anticoagulant treatment
- 3 patients had a post-traumatic pictures; specify the type of trauma
- Are all territories accessible to thrombectomy ?
- 2 patients are young (< 18 y.o.) ; was the inclusion of minors authorised by the ethics committee?

Author Response

- the 20 patients all have a severe CVST or refractory to treatment but a few patients had early thrombectomy (average delay: 4 days ; 0 to 21 days). What event decided the procedure in these patients with a severe initial picture ?  
I think the presentation was a bit misleading here, symptom onset is defined as the time the patient retrospectively first described symptoms (like first occurance of therapy refactory headache). Clinical worsening usually led to clinical admission and thrombectomy was performed within the first 36h after admission in the most cases (14/20). We added an additional column in table 3 to report the time between admission and thrombectomy and also added this within the result section.

Thrombectomy in stroke should be performed within the first few hours; why should this instruction not be valid for CVST ?
Isn't there a risk of reduced efficacy if the procedure is delayed ?                  
In arterial thrombectomy damage is irreversible within a very short time frame. CVST on the other hand has a much better prognosis even in severe cases (medical treatment alone 37% good clinical outcome in ischemic stroke, and 87% in CVST), and clinical use of venous thrombectomy is still under debate. In our experience it usually is justified to wait for a short period of time, to see of the clinical status shows a beneficial course or if a rapid further deterioration is seen. 
Nevertheless, if the procedure is delays the thrombus usually is hard to retrieve and we even have seen cases in which thrombectomy failed that were attributed to clot configuration over time. Usually we try to perform thrombectomy in severe cases as possible. Further prospective and retrospective studies are required, to identify patients at a high risk of developing a unfavourable outcome and also to evaluate the outcome with and without venous thrombectomy in such a cohort. 

How many patients had the thrombectomy on the day of diagnosis (D0), what was the outcome for these patients ?
Please see the new Table 3 for these results. Day of admission was the day of diagnosis, in 7 patients thrombectomy was performed at the same day, and in additional 7 patients within the first 24 - 36h at the following day. 

- regarding the anticoagulant treatment, the data are insufficient: type of heparin (UFH and/or LMWH), dosage, biological results (APTT ratio, anti-Xa activity)? definition of resistance to anticoagulant treatment

We agree that this information is relevant, but it is not available as the longitduinal laboratory data was not within the dataset requested and approved by the ethics committee. All data was anonymized after extraction of the defined clinical parameters in accordance with the local data protection law, that allows an evaluation of clinical data without permission of the individual patient but requires an anonymization as soon as possible as described in the methods section. We added the missing biological results to the limitations section.

Clinical standard in the early treatment phase is usually UFH i.v. (this was clarified in the methods section) with a defined PTT range between 60 and 80s in cases with traumatic brain injury usually >40s, which is monitored clinically and by imaging (as described in the methods section). Increasing thrombus size, new thrombosed sinus or appearance or progressive intracranial hemmorrhage is usually considered restistance to anticoagulant treatment. This has been described in the methods section as "Refractory CVST was defined as deterioration of clinical status despite medical man-agement,..."

- 3 patients had a post-traumatic pictures; specify the type of trauma
All three patients had a skull fracture that crossed a transvers sinus, with traumatic sinus thrombosis and additional traumatic intracranial hematomas (see table 2). This was clarified within the text.

- Are all territories accessible to thrombectomy ?
All venous sinus are usually accessible (this was added in the methods section), while bridging veins, deep veins and cortical veins are not accessible (due to the very high risk of rupture), as described in the limitations: "... if the thrombus extends from the sinus into these small veins which MT is unable to recanalize because of the risk of perforation [30]."

- 2 patients are young (< 18 y.o.) ; was the inclusion of minors authorised by the ethics committee?
Inclusion of all patients that underwent mechanical thrombectomy in a 10 year time period was authorised by the ethics committee, irrespective of age. But after extraction of the defined clinical data, reidentification lists had to be destroyed (anonymized dataset).

Reviewer 2 Report

A very interesting study demonstrating the possible beneficial effect of mechanical thrombectomy in refractory cases of cerebral venous thrombosis.

As is already known, CVT represents a rather rare pathology, or more precisely an under-diagnosed pathology. The data obtained are from a single center, and their accuracy may be influenced by the experience of that center with mechanical thrombectomy in CVT.

The design of the study is very interesting, the methods used are clearly described; the author clearly explains the methods used and the inclusion criteria.

For the center where I work, this pathology represents a point of interest and I believe that such prospective or retrospective studies from other centers would be beneficial in order to support what this study shows us.

Otherwise, the discussions and the conclusion of this study correctly presented. 

Author Response

We thank the reviewer for his comments.